# Nutrition, Gut Microbiota, and Allergy Development in Infants

**DOI:** 10.3390/nu14204316

**Published:** 2022-10-15

**Authors:** Alejandra Rey-Mariño, M. Pilar Francino

**Affiliations:** 1Genomics and Health Department, Foundation for the Promotion of Health and Biomedical Research of the Valencia Region (FISABIO), 46020 València, Spain; 2CIBER en Epidemiología y Salud Pública (CIBERESP), 28001 Madrid, Spain

**Keywords:** gut microbiome, infancy, microbial succession, metagenomics, metatranscriptomics, early nutrition, breast milk, atopy, allergies, food allergy

## Abstract

The process of gut microbiota development in infants is currently being challenged by numerous factors associated with the contemporary lifestyle, including diet. A thorough understanding of all aspects of microbiota development will be necessary for engineering strategies that can modulate it in a beneficial direction. The long-term consequences for human development and health of alterations in the succession pattern that forms the gut microbiota are just beginning to be explored and require much further investigation. Nevertheless, it is clear that gut microbiota development in infancy bears strong associations with the risk for allergic disease. A useful understanding of microbial succession in the gut of infants needs to reveal not only changes in taxonomic composition but also the development of functional capacities through time and how these are related to diet and various environmental factors. Metagenomic and metatranscriptomic studies have started to produce insights into the trends of functional repertoire and gene expression change within the first year after birth. This understanding is critical as during this period the most substantial development of the gut microbiota takes place and the relations between gut microbes and host immunity are established. However, further research needs to focus on the impact of diet on these changes and on how diet can be used to counteract the challenges posed by modern lifestyles to microbiota development and reduce the risk of allergic disease.

## 1. Introduction

In recent years it has become apparent that the adequate establishment of a balanced gut microbiota during early life is crucial for numerous aspects of human development and health. In particular, it has long been recognized that there is a link between gut microbiota development and allergy risk, supported by both epidemiological and clinical data [1]. Early alterations in the establishment of the gut microbial community can also have severe long-term consequences on other aspects of immune, metabolic, and neurological development that will need to be carefully investigated in the years to come [1,2,3,4]. This will require a solid understanding of all the dynamic processes that converge into gut microbiota development during infancy. Although the gut microbiota is not completely settled by the end of infancy, the first year after birth is the one that sees the most intense changes in the gut and its bacterial community, as it transitions from an environment with few microbes to a complex and densely populated ecosystem. Key events during this period that substantially affect gut microbiota composition include the introduction of solid foods to the infant’s diet and the cessation of breastfeeding [5,6,7]. Recent work has shed light on the progression of these changes, and, in particular, metagenomic and metatranscriptomic studies have revealed new insights into the early metabolic development of the gut microbial community. An understanding of microbial succession in the gut of healthy infants at different levels, encompassing not only taxonomic composition but also functional repertoire and gene expression changes, will be crucial for developing strategies aimed at modulating this process when altered health conditions or environmental exposures threaten to derail its normal course.

## 2. Taxonomic Succession in the Intestinal Tract of Healthy Infants

The process of gut microbiota development in infants has been most extensively surveyed at the level of taxonomic succession by means of culture or molecular analyses based on the 16S rRNA gene [5,6,7,8,9,10,11]. Such taxonomy-centered studies have been able to reveal important facts regarding the assemblage of the gut bacterial community. They have shown that the process of early gut colonization is extremely variable among individuals and is influenced by numerous factors. Among these, the mode of birth (i.e., vaginal vs. C-section) and the type of feeding (i.e., breast milk vs. formula) are known to be the main factors influencing the richness, diversity, and composition of the infant’s gut microbial community [5,8,12,13]. Inter-individual variability is most extensive during the earliest stages of the infant’s gut microbiota development, characterized by a relatively low taxonomic richness and a very uneven distribution of taxa within individuals. As infant development progresses, the microbial assemblages in different individuals tend to converge toward an adult-like composition with higher richness and a more even taxon distribution [6,7,10]. Succession is, however, not complete by the one-year mark, as significant differences remain between one-year-olds and their mothers in terms of taxonomic microbiota composition, richness, and diversity [14].

If we focus on what specific bacterial groups are present at different time points and how they change in abundance during microbiota development, we can see that at birth the infant’s meconium often contains mainly bacteria belonging to the Enterobacteriaceae and to Firmicutes genera such as *Bacillus*, *Enterococcus*, *Streptococcus*, and *Staphylococcus*, although *Bifidobacterium* and many other genera can also be found [15,16,17]. Meconium is formed by material accumulated in the fetal intestinal tract in utero, and the presence of these bacteria suggests that a low level of bacterial colonization has already occurred before birth. Nevertheless, the moment of birth clearly represents a major exposure to novel microbes, and, importantly, this exposure will be radically different if the infant is born by vaginal delivery or by C-section. In the first case, passage through the birth canal exposes the infant to its mother’s vaginal and fecal microbiota, including genera such as *Lactobacillus*, *Prevotella*, and *Sneathia*; in contrast, for infants born by C-section, the first encounter with bacteria outside of the womb is likely to be with skin bacteria present in the surrounding environment, such as *Staphylococcus*, *Corynebacterium*, and *Propionibacterium* [12]. Most vaginal and skin bacteria do not seem to take hold in the infant gut; however, their presence will likely affect the colonization capacities of other bacteria [18]. These priority effects may then result in long-lasting effects on microbiota assembly and on the various host processes that are affected by it, including immune, metabolic, and neurologic development [19]. Accordingly, infants delivered by C-section display alterations in the diversity and composition of their gut microbiota during the months after birth, although most compositional differences between C-section and vaginally delivered infants seem to disappear by one year of age [20]. Notably, infants born by C-section undergo delayed colonization by *Bifidobacterium* and *Bacteroides*, whereas they often harbor an elevated presence of *Clostridium difficile* [8,13,21,22,23,24,25].

During the first months of life, infants will be further exposed to a variety of microorganisms from different sources, some of which will be able to colonize their intestinal tract. Breast milk is a privileged source of microbes, as not only does it contain bacteria known to be beneficial to the infant, such as *Bifidobacterium*, *Lactobacillus*, and other lactic acid bacteria [26,27,28], but it also transfers immunosuppressive and anti-inflammatory cytokines that likely facilitate the establishment of tolerance to these bacteria [29]. Moreover, breast milk also provides nutritional components that select for the growth of bifidobacteria, mainly a variety of oligosaccharides acting, in fact, as natural prebiotics [30,31]. In contrast, cow’s milk contains a low amount of oligosaccharides, which, overall, display a limited structural variation in comparison to those in human milk. Consequently, *Bifidobacterium*-dominated microbiotas are more frequent among breastfed infants than among infants fed with formula, which tend to have a higher species richness with an over-representation of *Clostridium difficile* [13,32].

The introduction of solid foods into the diet is accompanied by an important shift in gut microbiota composition [6,7,14]. This shift entails an increase in abundance for numerous taxa that are found to be dominant in the adult gut microbiota, such as *Bacteroides* and *Ruminococcus*, and a concomitant decrease in *Escherichia* and other enterics. By one year of age, the gut microbiota has undergone a substantial increase in many Firmicutes genera, particularly the main butyrate producers in the gut, and a decrease in *Bifidobacterium*, which is likely dependent on the reduction or cessation of breast milk intake [33]. These changes result in a community dominated by the phyla Bacteroidetes and Firmicutes, mainly represented by the genera *Bacteroides*, *Faecalibacterium*, *Clostridium*, and *Ruminococcus* [14]. This microbiota composition already substantially resembles that of the adult, although microbiota maturation will continue until children are at least three years old [34], and, in fact, recent work suggests that a typical adult pattern may not be totally established until adolescence [35].

## 3. Metagenomic Analysis of Succession: Evolution of the Gut Microbiota Functional Repertoire

Although taxonomy-centered studies have clearly established how gut microbiota composition varies through time during infancy and among individual infants, the effect of this variability on the functional capacities of the community has yet to be explored to the same extent. In adults, metagenomic studies indicate that, in contrast to taxonomic variability, there seem to be conserved functional profiles among the microbiotas of different individuals [36]. This is consistent with the functional equivalence hypothesis, which, within the context of the neutral theory of community ecology [37], proposes that multiple species may possess similar functional attributes. Therefore, the taxonomically different assemblages in the gut of different healthy individuals could have little or no impact on the host.

The functional repertoire of the infant’s gut microbiota has been investigated in several metagenomic cross-sectional [34,38] and longitudinal studies [9,14,33,39,40]. These studies have indicated that, in spite of the taxonomic disparities, the functional capabilities of the microbiota in infants broadly mirror those of adults from very early on, with a gene repertoire dominated by carbohydrate metabolism functions. Nevertheless, the functional repertoire of the infant does change throughout the first year of life. Genes related to the tricarboxylic acid cycle are enriched in neonates, in accordance with the higher levels of oxygen in their gut environment. Additionally, the microbiota of the first months of life is enriched in genes required for the uptake and degradation of sugars present in breast milk, such as lactose and the various components of human milk oligosaccharides (HMOs), whereas the introduction of solid foods promotes enrichment in genes coding for the utilization of a larger variety of carbohydrates, including complex sugars and starch. In addition, solid food introduction changes the provision of ingested amino acids and vitamins to the infant, and the repertoire of bacterial genes related to amino acid and vitamin transport and biosynthesis is also found to change. Amino acid transport systems are enriched during the milk-feeding period, whereas many amino acid biosynthesis pathways tend to increase with time. Genes required for the synthesis of several vitamins, such as K2, B6, B9, and retinol are enriched in newborns, whereas those involved in the synthesis of vitamins B1, B5, and B12 increase later on. Interestingly, vitamin B6 has been identified as one of only a few vitamin deficiencies in mothers that lead to suboptimal concentrations in milk and adversely affect the infant [41,42].

Moreover, studies that have simultaneously analyzed the taxonomic and functional development of the gut microbiota during infancy have shown that individual instances of gut colonization vary at both levels in their temporal dynamics and that clear parallelisms exist between functional and taxonomic change. Therefore, the observed inter-individual variability in taxonomic composition during succession is not fully compensated by functional equivalence among bacterial genera and does influence the functional capacities of the microbiota. Consequently, successional variability may have important consequences, affecting host physiology, metabolism, and immunity, although, overall, a strong common pattern of directional change toward the functional composition of the adult microbiota can be detected across healthy infants [14].

Metagenomic studies also enable a more detailed analysis of the metabolic capacities available to the microbial community through the reconstruction of metabolic networks. The adaptation of the human gut microbiota metabolic network during the first year after birth has been analyzed by means of the metabolic reconstruction of microbiota-wide networks for different time points based on metagenomic data and nutrient input from food-frequency questionnaires [43]. This study detected significant alterations to the metabolic network once solid food is introduced into the diet, leading to a different pattern of output metabolites that can be potentially released from the gut microbiota to the host. For instance, vitamin B6 metabolism is more highly represented in the infant’s metabolic network before the solid diet introduction. On the other hand, the metabolism of phenolic compounds becomes highly represented in the infant’s metabolic network only after the introduction of solid food, with the activation of pathways for the biosynthesis of flavonoids and phenylpropanoids that employ precursors present in fruits and vegetables.

## 4. Metatranscriptomics: Stage-Specific Adaptation of Gene Expression in the Gut Microbiota of Infants

Metatranscriptomic studies are crucial to the understanding of the functioning of a microbial community as they can reveal substantial aspects of its biology that remain obscured when gene expression is not taken into account [44,45,46]. However, to date, a single metatranscriptomic study of the gut successional process has been performed [47]. This work has shown that, like the metagenome, the metatranscriptome of the infant changes throughout the first year of life, and that, in some cases, the temporal trends of gene expression differ from those of gene abundance.

In accordance with what is seen at the metagenome level, hallmarks of aerobic metabolism disappear from the microbial metatranscriptome of infants as development proceeds, while the expression of functions related to carbohydrate transport and metabolism increases and diversifies. A remarkable particularity of infants’ metabolism that is pinpointed by the metatranscriptomic analyses, but was not apparent in the metagenome, is related to butyrate biosynthesis. In adults, the butyrate produced by bacterial fermentation of carbohydrates in the gut is essential for the energy provision of colonocytes and the health of the colonic mucosa, as well as for many of the microbiota-mediated effects on metabolism and immunity. However, infants have a metabolic profile with very low proportions of butyrate in comparison to adults, as estimated by the concentration detected in stool [48,49]. In accordance, the common butyrate-producing bacteria of the adult gut, such as *Faecalibacterium* and *Eubacterium,* are rare in infants before the introduction of solid foods, while the *Bifidobacterium* and *Bacteroides* that dominate the gut microbiota during this period do not encode any pathway for butyrate synthesis. Therefore, it has been thought that the colonic mucosa of the young infant could have different requirements than those of the adult and that its enterocytes may use an alternative energy source. In contraposition, metatranscriptomics detected that some butyrate synthesis enzymes are overexpressed at three months of age, in particular those of the phosphotransbutyrylase pathway encoded by several *Clostridium* species that are often abundant in infants at this stage [50]. This suggests that butyrate production may be ensured in the gut of young infants by *Clostridium* species before the typical butyrate synthesizers of the adult gut become abundant [47].

By the end of the first year after birth, the infant’s metatranscriptome has notably increased in similarity to that of the adult, with the overexpression of many energy metabolism and carbohydrate transport functions. This likely reflects the diversification of the diet, as many infants consume a variety of products of plant and animal origin at this age. Nevertheless, numerous functions that are overexpressed in adults remain underexpressed in one-year-old infants, such as those related to methanogenesis; the binding, transport, and degradation of proteins; and regulatory protein interactions. This indicates that the metabolism of the microbiota still needs to further diversify after the infancy period. Remarkably, these same functions are also not upregulated in the maternal microbiota during late pregnancy, suggesting that late-coming functions that appear after the first year of life are not essential to the basic functioning of the gut microbiota and may be downgraded in specific physiological conditions. On the other hand, at one year the expression of transcription and translation functions is still higher than in adults, suggesting that the growth rate of bacterial populations may be faster in infants, in whom the microbial habitats of the gut may still not be saturated and the gut microbial community may not yet have reached a stable state [47]. The changes in gene expression patterns during infant development are illustrated in Figure 1, along with parallel changes in taxonomic composition and gene repertoire.

## 5. Implications of Early Gut Microbiota Colonization for Long-Term Health

During infancy, the interactions between host and gut microbes are established. Through these interactions, immune modulation and induction of immunological tolerance are produced, and, in turn, contribute to the correct further deployment of microbial colonization. Immune interactions with bacteria could begin in the uterus in the final stage of pregnancy with the transfer of intestinal bacteria from the mother to the fetus through their translocation to the mesenteric lymph nodes and from there to the systemic circulation [51,52,53]. Since these bacteria would be the first to be in contact with the infant, they could be marking immune and metabolic development; therefore, further research is necessary to understand the extent of microbial seeding in utero and its impact on long-term health.

Early microbial colonization patterns will have a strong impact on the maturation of the infant’s gut-associated lymphoid tissue (GALT). The development of GALT structures such as Peyer’s patches, mesenteric lymph nodes, and isolated lymphoid follicles responds to signals from the intestinal microbiota [54]. The gut microbiota interacts with the GALT through various mechanisms, including receptor molecules for ligands of microbial origin that are present in dendritic cells and a variety of other immune cells [55]. Among other things, binding of microbial ligands to these receptors causes the production of cytokines, which in turn initiate the differentiation of naïve T cells into regulatory cells (Tregs), which have an anti-inflammatory role [56,57,58], or into T helper (Th) cells, which play various key roles in the immune response [56,59]. In particular, Th2 responses involve the production of Immunoglobulin E (IgE), which activates the release of histamine by mast cells that results in inflammation and allergic symptomatology. Treg cells can inhibit the differentiation of naïve T cells into Th2 and other Th cell types [55,58], and an imbalance between Treg and Th cells, as a result of inadequate microbial colonization of the gut, can lead to a deregulation of the immune response toward various pathological states [60,61,62]. Experiments in germ-free mice have demonstrated the existence of a specific postnatal time window during which Tregs have to be exposed to microbial colonization in order to ensure the development of immunotolerance to later environmental exposures [63,64]. Furthermore, the gut microbiota stimulates the production of non-inflammatory IgA, which favors the development of immunotolerance [62,65]. However, different bacteria induce IgA production at different levels, and also seem to interact with this antibody in different ways. IgA has the capacity to engage in a variety of asymmetrical interactions with different bacterial types, potentially contributing to promoting the retention of habitual gut symbionts while ensuring containment or clearance of opportunistic pathogens [66]. For instance, interactions between IgA N-glycans and peptidoglycans in the cell wall envelope of beneficial Gram-positive species of the Firmicutes and Actinobacteria appear to favor their retention in the intestine. In contrast, bacteria of the Enterobacteriaceae family, which often behave as opportunistic pathogens, are usually strongly bound by IgA, and they overgrow in the intestine in cases of IgA deficiency [67], indicating that IgA contributes to their clearance or at least limits their growth. Therefore, microbial colonization patterns may both influence and be influenced by IgAs, and early colonizers may prime microbiota assembly partly through their impact on IgA production.

Consequently, the alterations of intestinal microbial colonization derived from the various environmental factors typical of industrialized societies that result in reduced exposure to microbes can contribute to the development of autoimmune and atopic diseases [1,68,69]. In particular, the development of T cell responses and the production of different cytokines have been shown to be impacted by C-section. Jakobsson et al. [24] demonstrated reduced Th1 responses during the first 2 years of life in infants born by C-section. The delayed Bacteroidetes colonization associated with C-section may be involved in this reduction, as *Bacteroides* species have been shown to induce the production of IL-10 and other cytokines that affect the Th1/Th2 balance and promote immunotolerance [70]. Similarly, Hansen et al. [71] showed that adult mice that had been born by C-section had fewer Tregs and tolerogenic dendritic cells in their mesenteric lymph nodes and spleens, as well as a lower IL-10 expression. Humoral antibody responses have also been shown to differ in infants born by C-section. Huurre et al. [22] showed that these infants have stronger humoral immune responses throughout the first year of life, with higher total numbers of immunoglobulin-secreting B cells in peripheral blood, suggesting decreased tolerance.

Clinical evidence also indicates that children born by C-section are at higher risk for several types of atopic diseases. In particular, several studies have associated C-sections with the risk for asthma, including two large meta-analyses that integrated 26 and 13 studies [72,73]. Furthermore, C-sections without ruptured membranes result in a 60% higher risk of asthma than those with ruptured membranes, likely due to lower exposure to microbes [74]. C-sections have also been associated with food allergy and allergic rhinitis [72], with food allergies being particularly elevated in children born by C-section to allergic mothers [75].

Several studies have also linked a greater exposure to microbes during childhood due to the presence of pets, siblings, domestic animals, or attending daycare, with a lower risk of suffering from atopic dermatitis, asthma, hay fever, and infectious diseases [76,77,78,79]. In the case of food allergy, various longitudinal or cross-sectional studies have demonstrated fewer challenge-proven or doctor-diagnosed cases in children with siblings or attending childcare before six months [80]. These findings highlight the close relationship between the microbial colonization of the intestine and the development of the immune system of infants, a balanced process that if altered can lead to the appearance of long-term diseases.

## 6. Breast Milk Feeding, Gut Microbiota, and Their Interactions with the Immune System

Breastfeeding plays a key role in the microbial colonization of the intestine and the development of the immune system in infants. Breast milk contains hundreds of non-pathogenic bacterial species, including *Bifidobacterium* and *Lactobacillus*, but also *Leuconostoc*, *Streptococcus*, *Enterococcus*, *Lactococcus*, *Weissella*, skin bacteria such as *Propionibacterium* and *Staphylococcus*, and even bacteria from the oral cavity, such as *Veillonella*, *Leptotrichia* or *Prevotella*, which increase along lactation [81,82]. These bacteria, upon reaching the infant’s intestine, train the immune system to differentiate between these habitual commensal microorganisms and those that are potentially harmful [83]. This is helped by the presence of several factors in breast milk such as immunoglobulins, glycoproteins, glycolipids, antimicrobial peptides, immunosuppressive and anti-inflammatory cytokines, and lymphocytes [29,84,85]. Various growth factors found in breast milk, such as hepatocyte, epidermal, and vascular endothelial growth factors, may also contribute to suppressing antigen sensitization [86]. For instance, murine models have shown that hepatocyte growth factor, which is found at very elevated concentrations in breast milk, suppresses the antigen-presenting capacity of dendritic cells [87]. HMOs have also been shown to promote allergen tolerance. In particular, 2′-fucosyllactose (2′-FL), which is the most abundant and prevalent HMO in breast milk, induces an increase in Treg populations in mouse models [88]. In agreement, studies have shown that formulas supplemented with this HMO promote lower levels of pro-inflammatory cytokines, similar to those present in breastfed infants [89].

In addition, as already mentioned, the HMOs in breast milk select for the growth of bifidobacteria, whose lack has been associated with systemic inflammation and immune dysregulation [90]. A mechanistic link between bifidobacteria and immunoregulation during the first months of life was established in a study that supplemented breast milk with *Bifidobacterium infantis* EVC001, a strain that expresses all HMO-utilization genes. In infants receiving this supplementation, intestinal Th2 and Th17 cytokines were silenced while interferon β was induced. Furthermore, their fecal water contained high levels of indolelactate produced by *B. infantis* that upregulated immunoregulatory galectin-1 in Th2 and Th17 cells [90].

In accordance with the various immunoregulatory capacities of breast milk, several studies have shown a lower incidence of infections, inflammatory diseases, and autoimmune diseases in breastfed infants compared to those who are formula-fed [91,92,93]. Regarding allergy, the protective role of breastfeeding is less clear, as different studies have generated conflicting results [85,94]. However, breastfeeding appears to show increased protective effects for infants with a family history of atopic disease and other risk factors [93]. Heterogeneity in study design and length of breast milk feeding surely contribute to the differences reported across studies. Various analyses have detected that breast milk feeding is effective in protecting against asthma, eczema, and allergic rhinitis when the recommendations of the World Health Organization are followed (i.e., exclusive breastfeeding during the first six months of life) [94,95]. Remarkably, the impact of breastfeeding is increased in low-income countries, suggesting that lifestyle-related differences in breast milk composition may contribute to unequal protective effects.

In this respect, various studies have compared breast milk in urban women and in Old Order Mennonites, a New York state population with a farming lifestyle, exposure to unpasteurized milk, and a low prevalence of allergic diseases. The breast milk composition was found to vary substantially, with a higher phylogenetic diversity of bacteria, a higher abundance of *Bifidobacterium infantis*, and higher levels of IgAs, TGF-β2, and IFN-λ3 in Old Order Mennonites [96,97]. These results are in agreement with previous studies showing that high levels of bifidobacteria in breast milk are associated with the secretion of IgA in the child’s gut, in addition to playing a key role in maintaining the mucosal barrier and providing anti-inflammatory effects [98]. Similarly, other studies have also shown that high levels of IgA and TGF-β correlate with a lower risk for atopic diseases in young children [92,99]. Furthermore, Holm et al. also identified three N-glycopeptides in breast milk secretory IgAs (SigAs) that had significantly higher abundances in Old Order Mennonite than in urban mothers, as well as higher abundances in non-atopic than in atopic mothers and children, irrespectively of the community in which they lived [100]. This result suggests that glycosylation of SIgAs is also an important factor affecting the protective effect of breast milk versus atopic disease, presumably by altering their affinity for particular microbes. Studies in these same populations also showed a strong correlation between maternal antibiotic use and the levels of several HMOs [97].

More generally, diet has been shown to have a great influence on the composition of breast milk. The fatty acid content of breast milk, derived from the diet, influences the composition of the infant’s cell membranes, which in the case of the cells of the immune system could affect their function [101], and could influence the development of immune diseases [91,102,103].

## 7. The Case of Food Allergies: A Protective Role for the Gut Microbiome

Studies on germ-free mice show that they are unable to develop immune tolerance to food antigens [104]. Moreover, when germ-free mice are colonized with the gut microbiome of sensitized susceptible mice, susceptibility to food antigens is also transferred [105]. In contrast, when they are colonized with microbiota from healthy donors, germ-free mice are protected against the development of cow’s milk allergy [106].

In agreement with such experimental data, several observational studies suggest that gut dysbiosis in early life can lead to the development of food allergies [107,108,109]. A recent study comparing twins with and without food allergies revealed differences in the composition of the gut microbiota and its metabolites, which suggests that the microbiota plays a key role in protecting against the development of this type of allergy [110]. The specific microorganisms involved in the development of food allergies and the manners in which microbes positively or negatively influence the tolerogenic mechanisms that prevent these allergies are not yet totally established, but recent works have made strides in this regard [111,112]. In particular, it has recently been shown that children affected by food or respiratory allergies present higher abundances of *Ruminococcus gnavus* and *Faecalibacterium prausnitzii,* as well as depletions of various species belonging to *Bifidobacterium, Bacteroides,* and a variety of fiber-degrading taxa [113]. Importantly, a metagenomic analysis revealed an increased pro-inflammatory potential in the microbiota of the allergic children, with enrichment of lipopolysaccharide- and urease-related genes. Moreover, within *R. gnavus*, strains differed between allergic and non-allergic children, with strains enriched in the former having a lower ability to degrade fiber, higher potential for adhering to the epithelium, and higher abundances of genes involved in the production of pro-inflammatory polysaccharides [113]. On the other hand, the abundance of *Alistipes* and *Bacteroides* has been related to dysbiosis in the intestine of children with non-IgE-related cow’s milk allergy compared to healthy children [114,115].

Furthermore, the composition of the maternal microbiota during pregnancy has also been related to the development of food allergies in the offspring. In particular, it has been shown that maternal carriage of *Prevotella copri* can protect from the development of IgE-mediated food allergy at 1 year of age [116]. The presence of *P. copri* may be particularly relevant as this species is known to occur at increased levels in non-Western populations with a low risk of allergic disease. It has been proposed that *P. copri* may protect from allergy via the production of succinate, which may promote the formation of dendritic cell precursors in the fetus [117].

In the intestine, there is continuous communication between the immune system and the metabolites derived from the microbiota. Short-chain fatty acids (SCFAs), in particular butyrate, seem to play a key role in preventing the development of allergic diseases, including food allergies [118]. SCFAs are derived from the fermentation of non-digestible dietary carbohydrates by the gut microbiota [119], and fecal levels of butyrate in early life have been related to the onset of allergic diseases [120]. In a cohort study of one-year-old children, high fecal butyrate levels were related to significantly less atopic sensitization to food and inhalant allergens and a lower incidence of asthma, food allergy, or allergic rhinitis between three and six years of age [121]. A longitudinal study in atopic children at three months and one year of age showed a correlation between the depletion of butyrate-producing bacteria and the development of allergic diseases later in life. Moreover, the gut microbiota of these children showed an under-representation of genes that encode key enzymes for butyrate production [122], confirming the protective role of butyrate in early life. Regarding the mechanisms through which butyrate may favor immune oral tolerance to food antigens, it has been seen in children with food allergies that butyrate shifts the pro-inflammatory state of monocytes toward an anti-inflammatory, regulatory and protective response state by enhancing the expression of IL-10, IFN-γ, and FOXP3, as well as promoting dendritic cells, Tregs, and the precursors of M2 macrophages [123]. In animal models, butyrate has been shown to reduce the appearance of allergic responses through the expression of tolerogenic cytokines and several biomarkers of gut barrier integrity [123].

## 8. Early Diet Impact on Food Allergy

The first two years of life are a very vulnerable period in which food allergies and other immune diseases can appear. During this time, several factors can influence the composition of the intestinal microbiota, its function, and the development of the immune system. Notably, breast milk could be a protective factor against the appearance of food allergies due to the numerous beneficial bacteria and bioactive factors that are present in it. Although the benefits of breastfeeding for preventing food allergies are not apparent in all studies, a recent analysis has shown that the breast milk of mothers whose children develop food allergies harbors a less abundant and species-rich microbiota than that of the mothers of healthy children [124]. Moreover, mothers of allergic children had lower abundances of *Bifidobacterium*, *Akkermansia*, *Clostridium* IV, *Clostridium* XIVa, *Veillonella*, and butyrate-producing bacteria such as *Fusobacterium*, *Roseburia*, and *Ruminococcus*, and higher abundances of Proteobacteria, in particular *Acinetobacter* and *Pseudomonas*. The lack of IL-13 in breast milk has also been associated with the development of food allergies [125].

In addition to containing beneficial bacteria, HMOs, and immune factors, breast milk is also an early source of butyrate, which, as explained above, plays a key role in the immunomodulation of the infant’s intestine. Moreover, the presence of dietary allergens in combination with immunoglobulins in breast milk may also foster their future toleration [126]. Therefore, exposure to common food proteins in a family’s diet through maternal breast milk may represent a protective factor against food allergies. In contrast, delayed exposure to allergen-containing foods in infants does not appear to be effective in allergy prevention [127].

With the introduction of foods other than breast milk or formula, the following patterns of diet diversification will inevitably influence the establishment of the child’s gut microbiota, being a key factor in the development of food allergies (Figure 2). The microbes contained in food may play an important role in this regard, as, for instance, consumption of unpasteurized cow’s milk is associated with a lower risk of developing allergies [128]. Several studies have also linked the consumption of a Mediterranean diet in children with a lower risk of suffering from food allergies and other allergic diseases, such as asthma and atopy [129,130,131]. This is probably due to this diet’s high content of non-digestible carbohydrates from the fiber contained in fruits, cereals, and legumes, which, when fermented by intestinal bacteria, produce SCFAs that protect against these allergic diseases [132]. On the contrary, the consumption in children and adolescents of a Western diet, characterized by a high content of fats and sugars and a low content of fiber, can result in a reduction in the levels of butyrate in the intestine and a higher incidence of allergic diseases [133,134,135].

## 9. Conclusions

Although individual patterns of gut colonization in healthy infants during the first year of life are enormously varied, some general trends have emerged that define microbial succession in this environment at the taxonomic, metagenomic, and metatranscriptomic levels (Figure 1). At each of these different levels, a directional pattern of change toward the adult state can be observed. However, temporal trends of gene expression do not always parallel those of gene abundance, as not all of the genes present in the microbiome at a given time will be expressed while rare genes may be upregulated. In fact, the relative expression of many genes changes through the first year of life indicate a stage-specific adaptation of gut microbiota activity during infant development. Importantly, significant differences remain between the gut microbiota of one-year-olds and that of adults at the levels of taxonomic richness, diversity, the complexity of interactions among taxa, gene repertoire, and gene expression patterns. This observation reinforces the notion that the gut microbiota assembly process is far from completed by the end of infancy, and that more research needs to be directed toward understanding the changes that occur during later periods such as childhood and adolescence.

Dysbiosis of an immature gut microbiota early in life can lead to negative health consequences even in adulthood due to its close interaction with the immune system. Butyrate and other SCFAs seem to play a key role in immunotolerance to food antigens, so breastfeeding and the consumption in childhood of a Mediterranean diet, rich in non-digestible carbohydrates that are transformed into butyrate in the colon, could reduce the risk of food allergies and other allergic diseases. In cases where breastfeeding is not possible, formula milk supplementation with prebiotics or probiotics could help balance the intestinal microbiota and serve as a preventative measure or a treatment to alleviate food allergies. However, further research into the mechanisms underlying the interplay between the gut microbiota and the immune system will be needed for the development of new approaches to treat allergic diseases.

## Figures and Tables

**Figure 1 nutrients-14-04316-f001:**
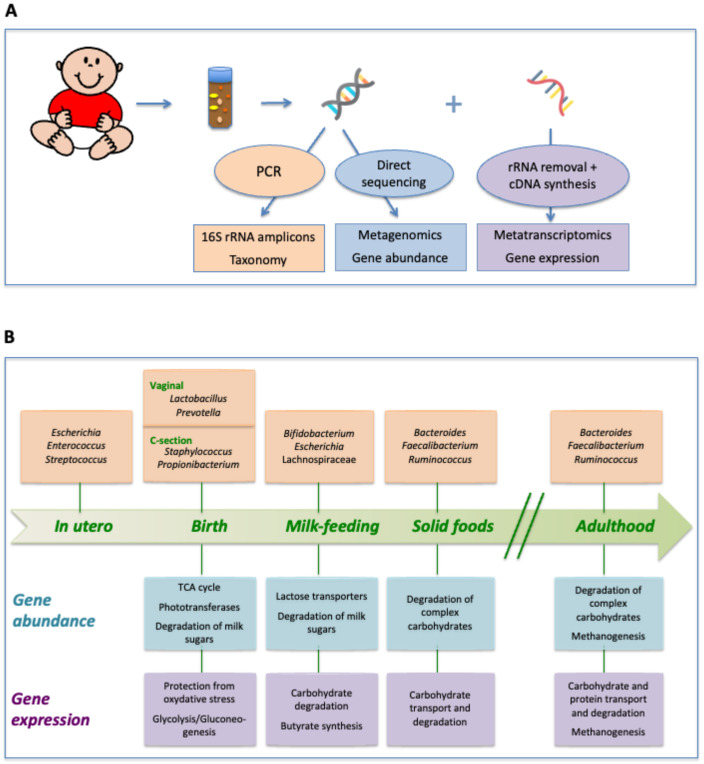
(**A**) DNA- and RNA-based omic approaches employed for the characterization of the infant gut microbiome. (**B**) Changes in taxonomic composition, gene abundance, and gene expression patterns during infant development.

**Figure 2 nutrients-14-04316-f002:**
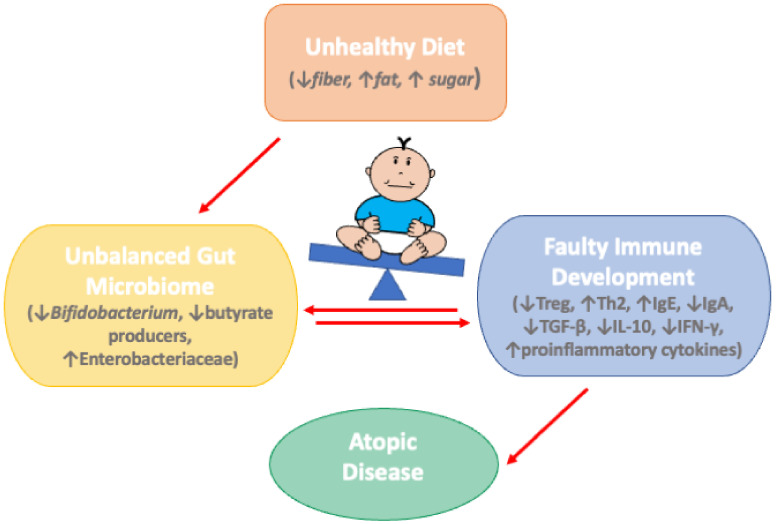
The effects of an unhealthy diet alter the development of the infant gut microbiota, resulting in faulty immune development and an increased risk of atopic disease.

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
