# Peer review of "Nutrition, Gut Microbiota, and Allergy Development in Infants"

_nutrients, 2022, doi:10.3390/nu14204316_

Round 1
Reviewer 1 Report
General writing feedback: Several very long sentences making the writing difficult to follow. Recommend shortening sentences where possible.
Abstract: Unclear what “this process” is on line 9. Sentence beginning on line 16 and ending on line 20 could be split into smaller sentences. Final sentence of abstract clearly defines purpose of review. Keywords are appropriate.
Introduction: Recommend including information about the links between the gut microbiome development and allergy in introduction, to highlight key purpose of review. Also recommend mentioning when the key gut microbiome changes occur during the first year of life up until three years of age, when the gut microbiome is considered “mature” (introduction of solid foods, cessation of breastfeeding etc.).
Taxonomic succession in the intestinal tract of healthy infants: Recommend splitting sentence beginning on line 53 and ending on line 57 into two sentences to highlight the change of uneven distribution of taxa in infancy to a more even distribution in adulthood. Suggest including information about when the gut microbiome converges to an adult pattern, which is considered to be three years of age (https://pubmed.ncbi.nlm.nih.gov/22699611/). When discussing the differences between delivery modes, consider mentioning that the gut microbiome differences gradually disappear after one year of age (https://www.nature.com/articles/s41467-019-13014-7).
Breast milk feeding, gut microbiota and their interactions with the immune system: Suggest removing the word “correct” from line 294, as a “normal” gut microbiome in infancy is difficult to describe. Consider comparing the diversity of gut microbiomes in breastfed vs. formula fed infants (breastfed infants generally have a lower overall diversity, but with more specialised gut microbiomes) (https://www.nature.com/articles/s41598-020-72635-x#:~:text=Studies%20have%20shown%20that%20infants,children10%2C11%2C12.).
Early diet impact on food allergy: Figure 2 is incomplete.
The potential role of the maternal microbiota in the development of infant allergy is not covered and should be for brevity. in particular the work fo Dr Peter Vuillermin has shown maternal carriage of prevotella copri protected from the development of allergy at 1 year of age. The authors should include a section with any studies on the topic of maternal microbiota in the development of offspring allergy.
Author Response
Pleae see the attachment.

Reviewer 2 Report
Dear Authors,
Thank you for a well written easy to ready review paper. A few comments though: Most importantly, please update Figure 2 on Page 10. Why are there parentheses and blank green oval?
Less importantly, this reviewers might have missed this study: https://www.nature.com/articles/s41467-021-26266-z
Recommend publishing when the above minor changes has been added.
Author Response
We thank the reviewer for the kind comments and for helping improve our manuscript. Our point-by-point answers follow and modified text is highlighted in yellow in the resubmitted manuscript.
- Most importantly, please update Figure 2 on Page 10. Why are there parentheses and blank green oval?
We apologize for the incompleteness of the figure, which was due to a format problem. The figure has been replaced by a tiff format, which should appear correctly.
- Less importantly, this reviewers might have missed this study:
https://www.nature.com/articles/s41467-021-26266-z
Thanks for pointing this study out to us. We have now cited it as reference number 117 and included it in lines 381-390 of the text: “In particular, it has recently been shown that children affected by food or respiratory allergies present higher abundances of Ruminococcus gnavus and Faecalibacterium prausnitzii, as well as depletions of various species belonging to Bifidobacterium, Bacteroides and a variety of fiber-degrading taxa [117]. Importantly, metagenomic analysis revealed an increased pro-inflammatory potential in the microbiota of the allergic children, with an enrichment of lipopolysaccharide- and urease-related genes. Moreover, within R. gnavus, strains differed between allergic and non-allergic children, with strains enriched in the former having lower ability to degrade fiber, higher potential for adhering to the epithelium and higher abundances of genes involved in the production of pro-inflammatory polysaccharides [117].”